# The Impact of Multidisciplinary Team Approach on Cytoreductive Surgery with Hyperthermic Intraperitoneal Chemotherapy for Peritoneal Carcinomatosis

**DOI:** 10.3390/jpm11121313

**Published:** 2021-12-06

**Authors:** Hao-Chien Hung, Po-Jung Hsu, Ting-Chang Chang, Hung-Hsueh Chou, Kuan-Gen Huang, Chyong-Huey Lai, Chao-Wei Lee, Ming-Chin Yu, Jeng-Fu You, Yu-Jen Hsu, Jun-Te Hsu, Ting-Jung Wu

**Affiliations:** 1Department of General Surgery, Chang-Gung Memorial Hospital at Linkou, College of Medicine, Chang-Gung University, Taoyuan 333, Taiwan; mp0616@cgmh.org.tw (H.-C.H.); mr1661@cgmh.org.tw (P.-J.H.); alanlee@cgmh.org.tw (C.-W.L.); a75159@cgmh.org.tw (M.-C.Y.); hsujt2813@cgmh.org.tw (J.-T.H.); 2Department of Obstetrics and Gynecology, Chang-Gung Memorial Hospital at Linkou, College of Medicine, Chang-Gung University, Taoyuan 333, Taiwan; tinchang@cgmh.org.tw (T.-C.C.); ma2012@cgmh.org.tw (H.-H.C.); kghuang@cgmh.org.tw (K.-G.H.); laich46@cgmh.org.tw (C.-H.L.); 3Department of Colon and Rectal Surgery, Chang-Gung Memorial Hospital at Linkou, College of Medicine, Chang-Gung University, Taoyuan 333, Taiwan; you3368@cgmh.org.tw (J.-F.Y.); m8295@cgmh.org.tw (Y.-J.H.)

**Keywords:** multidisciplinary team, cytoreductive surgery with hyperthermic intraperitoneal chemotherapy, peritoneal carcinomatosis, outcome

## Abstract

Background: Cytoreductive surgery with hyperthermic intraperitoneal chemotherapy (CRS–HIPEC) is a therapeutic approach used to achieve curative treatment in intra-abdominal malignancy with peritoneal carcinomatosis (PC). However, it is a complicated procedure with high post-operative complication rates. Thus, we analyzed our preliminary data to establish whether multidisciplinary teamwork (MDT) implementation is beneficial for CRS–HIPEC outcomes. Method: A series of 132 consecutive patients with synchronous or recurrent PC secondary to gastrointestinal or gynecologic cancer who received CRS–HIPEC operation between May 2015 and September 2017 were included. Ninety-nine patients were categorized into the MDT group, with the 33 other patients into the non-MDT group. Results: The mean PCI score was 16.3 ± 8.8. Patients in the MDT group more often presented a higher PCI score (*p* value = 0.038). Regarding CRS completeness (CCR 0–1), it was distributed 81.8% and 57.6% in the MDT and the non-MDT group, respectively (*p* value = 0.005). Although post-operative complications were common (*n* = 62, 47.0%), post-operative complication rates did not differ between the two groups. The cumulative OS survival rate at the first year was 75.5%. Older age (*p* = 0.030, HR = 4.58, 95% CI = 1.16–18.10), ECOG 2 (*p* = 0.030, HR = 6.41, 95% CI = 1.20–34.14), and incomplete cytoreduction (*p* = 0.048, HR = 2.79, 95% CI = 1.04–8.27) were independent prognostic factors for survival. Conclusions: Our experience suggests that the CRS–HIPEC performed under MDT cooperation may result in higher complete cytoreduction rates without increasing post-operative complications and hospital mortalities.

## 1. Introduction

Up to 5–20% of patients have been diagnosed with advanced intra-abdominal malignancies with peritoneal carcinomatosis (PC) at initial consultation due to lack of screening and a paucity of specific symptoms [1], with many suffering the consequences of a 6-months or less remaining lifespan [2]. In response, combination treatment using intravenous and intraperitoneal chemotherapy for microscopic residual cancer cell eradication following primary cytoreductive surgery for gross tumor removal has been shown to evidently prolong survival in selected PC patients [3,4,5,6].

For the past decades, in response to the complexity and diversification of cancer treatment, the role of the multidisciplinary team (MDT) has continually evolved in involving different professions to cooperate toward better therapeutic results, which has had a positive outcome [7,8]. However, only a few studies have focused on cytoreductive surgery-hyperthermic intraperitoneal chemotherapy (CRS–HIPEC), with none showing survival. Prior research from our allied institution, the Chiayi branch of the Chang-Gung Memorial Hospital, reported a higher complete cytoreduction rate and lower major complication rate of CRS–HIPEC under MDT guidance [9]. Therefore, we aimed to determine the impact of the MDT approach on CIRS–HIPEC effectiveness, safety, and outcomes.

## 2. Materials and Methods

### 2.1. Study Population and Study Design

Between May 2015 and September 2017, a series of 141 consecutive CRS–HIPEC procedures for PC secondary to miscellaneous primary malignancies in the Linkou Chang-Gung Memorial Hospital were retrospectively reviewed. All eligible patients were with an Eastern Cooperative Oncology Group (ECOG) performance of ≤2 with no evidence of extra-abdominal metastasis or bulky retroperitoneal extension, leading to be medically fit for cytoreductive surgery. Subsequent HIPEC procedures were then carried out following CRS [10]. Afterwards, we deducted patients who received repetitive CRS–HIPEC procedures (*n* = 4) and excluded those with a lack of essential data (*n* = 5), leaving a total of 132 patients for the study population.

A complete medical history, physical examination, routine preoperative hematology/biochemistry profiling, prior histopathology, computed tomography imaging, magnetic resonance imaging, or positron emission tomography (PET) of the chest, abdomen, and pelvis were reviewed prior to surgery.

### 2.2. Defining the Multidisciplinary Team Care with Application of CRS–HIPEC

Multidisciplinary peritoneal malignancy care is defined using integrated resources of core (mandatory) and extended (recommended) elements. The core membership in this approach includes the following: (1) an initiator, who is either a general surgeon, gyneco-oncologist, or proctologist, that starts the MDT approach before CRS–HIPEC procedure; (2) MDT coordinators, such as medical oncologists, imaging specialists, or pathologists; and (3) extended members, including anesthesiologists, thoracic surgeons, physiotherapists, psychologists, or social workers. Since surgical consultation would involve surgeons under two or more professional fields, MDT members altogether discuss individual patient cases at a given available time (whether facing each other, or via video or teleconferencing) to contribute professional counsel and suggestions on diagnosis and treatment. Additionally, a combined congress would be held to discuss and decide on procedure revisions ad-hoc depending on clinical findings based on procedure-related findings and post CRS–HIPEC outcomes. However, if the treatment process does not conform to defined regulations (e.g., the physician responsible does not seek an MDT approach or the MDT decision is discordant), then the case would be addressed as the non-MDT approach.

### 2.3. Cytoreductive Surgery and HIPEC Protocol

The sole purpose of curative-intent CRS is to diminish all macroscopically visible tumors, and thus aggressive resection of involved multiple affected visceral organs may be necessary, along with omentectomy and peritonectomy. Palliative CRS in the management of peritoneal carcinomatosis is mainly for patients with massive tumor burden or extensive abdominal cavity involvement, or predictable incomplete cytoreduction. Prophylactic HIPEC applies for non-metastatic cancer patients at high risk of tumor recurrence after curative-intent surgery. The HIPEC procedure is conducted using a closed method with a Performer^TM^ HT (RanD Biotech, Medolla, Italy). After temporary abdominal wound closure, four drains with fiberoptic temperature probes, including two inflow (lower horizon) and two outflow (higher horizon) catheters, were positioned and connected to a securely sealed extraperitoneal sterile circuit, which heats the perfusate constantly to hold an inflow temperature between 44–46 °C with an outflow temperature of at least 41–42 °C. This circulation initiates at a flow rate of 1000 mL/min, which is maintained for 30, 60, 90, or 120 min depending on the surgeon’s request, and the chemotherapeutic regimens are selected based on the primary malignancy. After circulation completion, the perfusate is drained out, and the abdominal cavity was rinsed with large amounts of cold normal saline. The wound was then re-opened to inspect the intra-abdominal status and restore bowel continuity if needed via reconstruction or enterostomy in order to avoid hazardous anastomosis.

Post-CRS–HIPEC ICU observation is not routinely applied, except for those fulfilling these selective criteria: ECOG status (=2), old age (≥65), excessive blood loss (≥1000 mL), advanced CRS extent/multiple organ resection (≥3), prolonged operation time (≥12 h), and hemodynamic instability during operation/advice from anesthesiologist based on previous study [11].

### 2.4. Post-Operative Complications and Outcome Surveillance

The Clavien–Dindo Classification system is used to grade surgical morbidity and mortality, wherein a minor post-operative complication is defined as less than or equal to Grade II complications, and surgical mortality is defined as Grade V (Death) if detected within 30 days after the operation or during hospitalization, regardless of the cause.

After discharge, all patients were followed up at the outpatient clinic once every three months with a full physical examination. Imaging studies would be obtained if it is indicated on symptomatic, clinical or serological evaluation. If necessary and clinically applicable, post-CRS–HIPEC systemic chemotherapy can also be arranged by medical oncologists.

### 2.5. Data Forms and Statistical Analysis

Mean values ± standard deviations with minimum and maximum ranges were used for continuous and categorical variables listed with numbers (percentage). Pearson’s chi-square test and independent T-test were used to compare clinical parameters.

Peritoneal malignancy was investigated using the peritoneal cancer index (PCI), wherein Grade I presents for a score of <9, II for 10–19, III for 20–29, and IV for 30–39 following the Sugarbaker classification [12]. Regarding cancer resection (CCR) completeness after CRS, the grade was evaluated by a surgeon at the end of procedure and was documented into four categories: CCR-0, no macroscopic residual cancer; CCR-1, for residual nodules < 2.5 mm; CCR-2, for residual nodules between 2.5 mm and 25 mm; CCR-3, for residual nodules > 25 mm [12]. In this case, complete CCR was defined as CCR 0–1.

Our primary endpoint was the post-operative complication occurrences, with overall survival after operation as the second endpoint. Variables that seemed to be associated with our investigated endpoints were evaluated in a multivariate analysis using the Cox proportional hazards regression model; however, the MDT approach was unconditionally selected due to its decisive role in the present study. Survival was measured from the CRS–HIPEC procedure date until death or latest follow-up, using Kaplan–Meier methods to assess the differences between subgroups by log-rank test. All *p* values were two-tailed, and a significant difference was determined at <0.05. The SPSS version 24.0 (IBM corporation, Armonk, NY, USA) software was used for all analyses.

## 3. Results

### 3.1. Population Composition

This study enrolled a total of 132 patients who underwent CRS–HIPEC procedure with primary diagnoses of ovarian cancer (*n* = 60), colorectal cancer (*n* = 25), appendiceal cancer (*n* = 10), gastric cancer (*n* = 12), and other primary malignancies (*n* = 25), 6 for biliary cancer, 5 for uterus or uterine tube cancer, 3 for mesothelioma, 3 for pancreatic cancer, 3 for small bowel cancer, 2 for retroperitoneal malignancy, 1 for bladder cancer, 1 for prostate cancer, and 1 for primitive neural ectoderm malignancy). Moreover, 68 (51.5%) patients had recurrent diseases after primary curative-intent surgical resection. The association between CRS–HIPEC operations and procedure initiators was investigated, showing that 60 cases were performed under the guidance of the general surgeons, while 56 and 16 cases were led by gyneco-oncologists and proctologists, respectively.

Among all patients, there were 35 males and 97 females, with a mean age of 54.6 years (range: 27.1–84.1) and a mean PCI of 16.3 ± 8.8. Only 14 patients (10.6%) were classified into ECOG 2, whereas 60 (45.5%) and 58 (43.9%) patients were in ECOG 0 and 1, respectively. Curative-intent CRS–HIPEC procedures were performed for 111 (84.1%) patients, palliative operations were carried out for 18 (13.6%; 3 ovarian cancer, 4 colorectal cancer, 2 appendiceal cancer, and 4 with other primary malignancies) patients, and 3 (2.3%; 2 gastric cancer and 1 gallbladder cancer) cases underwent prophylactic operations.

### 3.2. Demographic and Clinical Characteristics of the Patients

We further divided the enrolled patients into two main groups based on whether the MDT approach was applied or not (MDT, *n* = 99; non-MDT, *n* = 33; Table 1), showing no significant differences in age, gender, and baseline ECOG between the two groups. In contrast, a higher proportion of diabetes and hypertension seemed to be associated with the non-MDT group (*p* value = 0.024 and 0.002, respectively), while no significant results were found regarding the other comorbidities. Moreover, MDT group patients more often presented higher PCI scores (*p* value = 0.038), but there was no significant difference in cancer types or clinical symptom severity.

### 3.3. CRS–HIPEC Procedures and Intents

The average number of resected affected organs was higher in the MDT group (6.6 ± 3.4) as compared to the non-MDT group (3.5 ± 2.7) (*p* value < 0.001), with longer operation times, higher estimated blood loss, and increased need in blood transfusion in the MDT group (*p* value = 0.002, 0.023, 0.008, respectively). Regarding complete CRS (CCR0-1), it was distributed by 81.8% and 57.6% in the MDT and non-MDT groups, respectively (*p* value = 0.005). However, HIPEC settings were not significantly different between both groups in terms of surgical intention, HIPEC duration, HIPEC regimen digits, and averaged in/out flow temperatures (Table 2).

Regarding palliative-intent HIPEC, the mean PCI score was 22.2 ± 9.8, which was higher compared to the mean PCI score of 15.6 ± 8.0 in curative-intent HIPEC patients (*p* < 0.001). Moreover, patients who underwent palliative-intent HIPEC had less proportion of multi-visceral resections (66.7% vs. 91.9%; *p* = 0.002) and CRS completeness (16.7% vs. 83.8%; *p* < 0.001) than patients who received curative-intent HIPEC, subsequently leading to significantly inferior outcomes (*p* = 0.027, log-rank).

### 3.4. Post-Operative Complications

Short-term outcomes in this study included hospital stay length, ICU admission/stay rate, and ventilator use duration, as detailed in Table 3. For enrolled patients after CRS–HIPEC, 62 (47.0%) encountered post-operative complications, with nearly 60% facing no or minor complications in both groups. Six cases (4.5%) were surgical mortalities, and the remaining complications did not differ between the MDT and non-MDT groups. Concerning the six hospital mortalities, cardiopulmonary failure accounted for 50% of them (two cases with acute myocardial infarction, and the other two with severe pneumonia), followed by two cases of acute kidney injury. We also disclosed that post-surgical bowel perforation or anastomosis insufficiency was highly associated with major complications (*p* < 0.001) but not hospital mortalities (*p* = 0.553); however, protective enterostoma creation did not decrease bowel perforation or leakage rates (*p* = 0.687).

### 3.5. Univariate and Multivariate Analyses of Post-Operative Complication Predictions

To evaluate and identify independent risks for adverse post-operative major complications in the present study, we conducted Cox regression analysis (Table 4). This indicated that longer operation time > 12 h (*p* = 0.011, hazard ratio [HR] = 3.54, 95% CI = 1.33–9.43), complete cytoreduction (*p* = 0.035, HR = 3.48, 95% CI = 1.09–11.05), and gastrointestinal tract reconstruction (*p* = 0.047, HR = 2.58, 95% CI = 1.01–6.55) were independently influential.

### 3.6. Independent Survival Prognostic Factors

Survival prognostic factors are summarized in Table 5. In univariate analysis, several factors, including older age, male gender, other ovarian cancers, ECOG > 0, incomplete CCR, and post-operative complications, demonstrated inferior outcomes. In multivariate analysis, the three identified prognostic factors were older age (*p* = 0.030, HR = 4.58, 95% CI = 1.16–18.10), ECOG 2 (*p* = 0.030, HR = 6.41, 95% CI = 1.20–34.14), and incomplete cytoreduction (*p* = 0.048, HR = 2.79, 95% CI = 1.04–8.27).

### 3.7. Comparison of Post-CRS–HIPEC Survival

After a median follow-up of 4.5 months (6.8 ± 6.8) for the 132 patients following CRS–HIPEC, 26 (19.7%) of them died, with over half of the living patients (*n* = 70, 53.0%) reporting no obvious disease recurrence during the follow-up period. For all enrolled patients, the cumulative OS survival rate at the first year was 75.5%.

In groups between different ages, the 1-year-OS was 80.0% and 41.4%, respectively (*p* value = 0.001; Figure 1A). To compare with 65.8% of one-year-OS in male group, the female group deliberated a superior survival of 79.3% (*p* value = 0.036; Figure 1B). Moreover, a leading OS at one year of 86.4% in the ovarian cancer group was significantly dominant compared to other malignancies, although there was no difference between the remaining groups (*p* values; ovarian vs. colorectal: 0.039, ovarian vs. appendiceal: 0.007, ovarian vs. gastric: < 0.001, ovarian vs. others: 0.013; Figure 1C). Additionally, poorer ECOG showed worse outcomes in the progressive process (*p* value; ECOG 0 vs. ECOG 1: 0.012, ECOG 0 vs. ECOG 2: < 0.001, ECOG 1 vs. ECOG 2: 0.072; Figure 1D). As for cytoreduction completeness, the complete CCR group significantly exceeded the incomplete group by 82.8%–54.6% at 1-year-OS (*p* value < 0.001; Figure 1E). Furthermore, the absence of post-operative complications over-ranked the minor and major grade groups by 87.7%–67.4% and 55.7% at 1-year-OS, respectively (*p* value = 0.060 and 0.007, respectively; Figure 1F).

The following factors were calculated in UV: gender, cancer types, ECOG, smoke, alcohol, diabetes, hypertension, abdominal operation history, co-malignancy, viral hepatitis, heart disease, pre-CRS CT, severity of clinical symptoms, operation method, blood loss amount, intraoperative blood transfusion, multi-visceral resection, and creation of enterostomy; only significant results are shown in this table.

## 4. Discussion

Our MDT has grown to include the core membership of an initiator, medical oncologists, imaging specialists, pathologists, and a coordinator, as well as extended members of anesthesiologists, thoracic surgeons, physiotherapists, psychologists, and social workers as directed under comprehensive patient-centered care. In addition to the inter-MDT support system, which includes operational consultation, functional communication, and a clear definition of roles within the team members, a systemic review emphasized the importance of leadership as essential to MDT quality [13]. Our CRS–HIPEC MDT team initiator could belong to a subspecialty, which presents as the domination of the CRS process rather than adopting a subspecialist team in particular cancer. The concept is to treat all peritoneal cancers by the most advantageous MDT team approach, and to bring together the range of specialists is nevertheless a necessity for a comprehensive treatment plan for not only intraoperative assistance but also postoperative adjuvant therapy. A requirement for multiple involved organ resection with or without reconstruction is a successful CRS, with more subspecialized and more well-trained surgeons handling such situations within the bounds of their specialized surgical profession. Subspecialized team members are helpful to resolve the centralized power of one single surgical team.

A predominant role of CRS–HIPEC in treating peritoneal malignancies has been established in previous decades; however, high surgical morbidity and mortality remain major obstacles [14,15]. The major complication rate after CRS–HIPEC has been reported to go beyond 60% [16], and the surgical mortality rate has been reported to range from 0.9 to 5.8% [17]. In the present study, there were 22 (16.7%) pneumonia cases and 18 (13.6%) intra-abdominal infection cases among the 24 (18.2%) patients who needed intensive care or intervention under either local or general anesthesia, whereas 6 (4.5%) patients died following the CRS–HIPEC procedure. Prolonged ileus with digestive dysfunction was also reported in 30 patients after CRS–HIPEC during hospitalization, which was defined as immobilized bowel movement accompanied with enteral feeding intolerability that required total parenteral or prolonged partial parenteral nutrition for more than seven days. This condition could possibly be attributed to prolonged digestive function rehabilitation from pre-existing bowel obstruction and excessive intraoperative bowel manipulation as a result of intra-abdominal infection or HIPEC-related bowel damage.

In the current study, a longer operation time of >12 h (*p* = 0.011, hazard ratio [HR] = 3.54, 95% CI = 1.33–9.43), complete cytoreduction (*p* = 0.035, HR = 3.48, 95% CI = 1.09–11.05), and gastrointestinal tract reconstruction (*p* = 0.047, HR = 2.58, 95% CI = 1.01–6.55) were independent post-operative complication risk factors. These findings affirm proposals from a prospective study [18], demonstrating that long operation time, intraoperative gastrointestinal tract exposure, chemotherapy-related immune-depressive status, post-operative intra-abdominal drainage, central venous route and urinary catheter usage, ICU transit, and prolonged hospital stay would affect post-operative complications. Aside from introducing prevention, surveillance, and treatment protocols, this study also emphasized a multidisciplinary approach, wherein surgeons, infectologists, microbiologists, and immunologists take part in minimizing adverse events, ultimately leading to zero short-term mortalities.

Moreover, the previous study reported several independent risk factors for major morbidities, including prior extensive surgery, recent smoking behavior, poor physical performance status, and extensive cytoreduction [19]. Serial studies from Bakrin et al. also found a high correlation between the PCI score and post-operative complications [20,21,22]. Despite high PCI proportion and association of MDT with the number of involved resected organs, long operation time, intraoperative blood loss, and blood transfusion necessity, there was no significant difference in the complication incidence between the MDT and non-MDT groups. Nevertheless, an 81.8% of CCR 0–1 status with no or minimal residual disease achieved by MDT was a surprising finding.

An opportunity to explore how the comparative design of the MDT approach affected cytoreduction completion and helped the disease control process was also provided by our work. A high correlation between complete CRS (CCR 0–1) and multi-visceral resections (*p* = 0.007) was reported, with the MDT approach achieving 90.9% multi-visceral resections and 81.8% CRS completeness. This could possibly explain how the MDT approach improved cytoreduction completion in our study.

Concerning survival outcomes, there were three independent prognostic factors identified in multivariate analysis: older age, ECOG 2, and incomplete cytoreduction. Additionally, one prospective study revealed that higher PCI scores were associated with incomplete macroscopic cytoreductions other than CCR 0, which is a significant survival predictor, associated with worse time to disease progression and survival outcome [23]. More studies have also exhibited that complete cytoreduction indicated superior survival [24,25]. Of particular interest is the indicative role of MDT in mediating complete cytoreductive surgery for advanced cancerous conditions in this study. Although it did not appear as a prominent factor in survival analysis due to a significant PCI class mismatch between the two groups, we believe that an aggressive and comprehensive surgical intervention under meticulous operational planning from a variety of professions could be essential for patient benefits.

Despite these findings, this study has several limitations. It was retrospectively conducted by a single institution, and the median follow-up period was relatively short. Our results should be interpreted with caution for lack of discussions of several important prognostic factors, such as gene mutations, chemotherapy, and immunotherapy regimens. To avoid sample diversity by specific cancer type and standardize HIPEC protocols, prospective and large scale studies are required to draw a valid conclusion for these concerns.

## Figures and Tables

**Figure 1 jpm-11-01313-f001:**
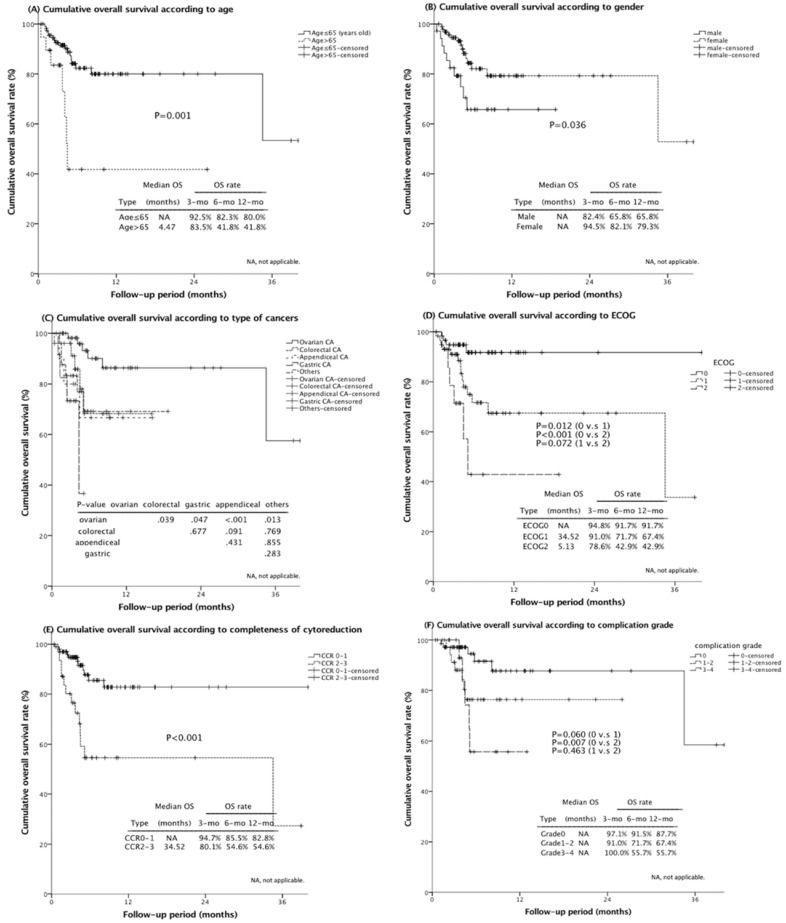
Kaplan–Meier plots of overall survival according to different subgroups. (**A**) Patients with older age (>65-year-old) demonstrated poor outcomes (*p* value = 0.001). (**B**) Female gender seemed to have a superior survival outcome (*p* value=0.036). (**C**) A more favorable outcome after CRS–HIPEC procedure was deliberated in patients with ovarian cancer than other malignancies (*p* value, ovarian vs. colorectal: 0.039; ovarian vs. appendiceal: 0.007; ovarian vs. gastric: <0.001; ovarian vs. other cancers: 0.013). (**D**) As the ECOG increases, the corresponding survival seemed to decrease (*p* value, ECOG 0 vs. ECOG 1: 0.012; ECOG 0 vs. ECOG 2: <0.001; ECOG 1 vs. ECOG 2: 0.072). (**E**) Patients achieved CCR 0-1 had better one-year OS than who did not (82.8% vs. 54.6%, *p* value < 0.001). (**F**) The presence of major postoperative complication was associated with a worse OS (absence vs. minor grade, *p* value = 0.060; absence vs. major grade, *p* value = 0.007).

**Table 1 jpm-11-01313-t001:** Demographic and clinical characteristics of the patients who underwent CRS–HIPEC.

	Non-MDT, N = 33	MDT, N = 99	*p*-Value
Basic conditions			
Age			NS
≤65	28 (84.8%)	85 (85.9%)	
>65	5 (15.2%)	14 (14.1%)	
Gender			NS
Male	13 (39.4%)	22 (22.2%)	
Female	20 (60.6%)	77 (77.8%)	
ECOG performance			NS
0	15 (45.5%)	45 (45.5%)	
1	13 (39.4%)	45 (45.5%)	
2	5 (15.1%)	9 (9.0%)	
Smoke			NS
No	28 (84.8%)	89 (89.9%)	
Yes	5 (15.2%)	10 (10.1%)	
Alcohol use			NS
No	29 (87.9%)	89 (89.9%)	
Yes	4 (12.1%)	10 (10.1%)	
Diabetes			0.024
No	23 (69.7%)	86 (86.9%)	
Yes	10 (30.3%)	13 (13.1%)	
Hypertension			0.002
No	17 (51.5%)	79 (79.8%)	
Yes	16 (48.5%)	20 (20.2%)	
Heart disease			NS
No	32 (97.0%)	97 (98.0%)	
Yes	1 (3.0%)	2 (2.0%)	
Co-malignancy			NS
No	30 (90.9%)	92 (92.9%)	
Yes	3 (9.1%)	7 (7.1%)	
Previous op			NS
No	16 (48.5%)	33 (33.3%)	
Yes	17 (51.5%)	66 (66.7%)	
Abdomen operation history			NS
No	15 (45.5%)	33 (33.3%)	
Yes	18 (54.5%)	66 (66.7%)	
Tumor factors			
Status			NS
Primary	20 (60.6%)	44 (44.4%)	
Recurrent	13 (39.4%)	55 (55.6%)	
Previous CT			NS
No	12 (36.4%)	29 (29.3%)	
Yes	21 (63.6%)	70 (70.7%)	
Cancer type			NS
A	9 (27.3%)	51 (51.5%)	
B	10 (30.3%)	15 (15.2%)	
C	1 (3.0%)	9 (9.1%)	
D	4 (12.1%)	8 (8.1%)	
E	9 (27.3%)	16 (16.2%)	
PCI class			0.038
I	14 (42.4%)	19 (19.2%)	
II	7 (21.2%)	42 (42.4%)	
III	9 (27.3%)	30 (30.3%)	
IV	3 (9.1%)	8 (8.1%)	
Clinical symptoms			NS
None	4 (12.1%)	11 (11.1%)	
Mild	21 (63.6%)	74 (74.7%)	
Severe	8 (24.3%)	14 (14.2%)	

Abbreviation: MDT, multidisciplinary team; NS, not significant; ECOG, Eastern Cooperative Oncology Group; PCI, peritoneal cancer index; op, operation; CT, chemotherapy; cancer type A, ovarian; cancer type B, colorectal; cancer type C, appendiceal; cancer type D, gastric; cancer type E, others; PCI class I, score 0–9; PCI class II, score 10–19; PCI class III, score 20–29; PCI class IV, score 30–39.

**Table 2 jpm-11-01313-t002:** Surgical settings and procedures of the patients who underwent CRS–HIPEC.

	Non-MDT, N = 33	MDT, N = 99	*p*-Value
Surgical characteristics			
Therapeutic initiator			0.001
GS	17 (51.5%)	44 (44.4%)	
GYN	7 (21.2%)	48 (48.5%)	
Proctologist	9 (27.3%)	7 (7.1%)	
Method			NS
Laparotomy	33 (100.0%)	95 (96.0%)	
Laproscopy	0 (0.0%)	4 (4.0%)	
Number of organs resected	3.5 ± 2.7	6.6 ± 3.4	<0.001
Multiple visceral resections			NS
No	6 (18.2%)	9 (9.1%)	
Yes	27 (81.8%)	90 (90.9%)	
Completeness of CRS			0.005
0–1	19 (57.6%)	81 (81.8%)	
2–3	14 (42.4%)	18 (18.2%)	
GIT reconstructive anastomosis			NS
No	15 (45.5%)	45 (45.5%)	
Yes	18 (54.5%)	54 (54.5%)	
Number of GIT anastomosis			NS
0	15 (45.5%)	44 (44.4%)	
1	11 (33.3%)	31 (31.3%)	
≥2	7 (21.2%)	24 (24.2%)	
Creation of enterostomy			0.049
No	30 (90.9%)	74 (74.7%)	
Yes	3 (9.1%)	25 (25.3%)	
Operation time, hours			0.002
≤12	31 (93.9%)	65 (65.7%)	
>12	2 (6.1%)	34 (34.3%)	
Blood loss, mL			0.023
≤500	28 (84.8%)	63 (63.6%)	
>500	5 (25.2%)	39 (26.4%)	
Blood transfusion			0.008
No	28 (84.8%)	59 (59.6%)	
Yes	5 (15.2%)	40 (40.4%)	
HIPEC settings			
Intent of HIPEC			0.003
Prophylatic	3 (9.1%)	0 (0.0%)	
Curative	23 (69.7%)	88 (88.9%)	
Palliative	7 (21.2%)	11 (11.1%)	
HIPEC duration, mins			NS
30	0 (0.0%)	1 (1.0%)	
60	11 (33.3%)	18 (18.2%)	
90	22 (66.7%)	77 (77.8%)	
120	0 (0.0%)	3 (3.0%)	
HIPEC regimen			NS
Single	10 (30.3%)	16 (16.2%)	
Dual	22 (66.7%)	80 (80.8%)	
Triple	1 (3.0%)	3 (3.0%)	
Inlet temperature, °C	43.6 ± 0.8	43.4 ± 0.8	NS
Outlet temperature, °C	41.8 ± 0.7	41.7 ± 0.6	NS

Abbreviation: GS, general surgeon; GYN, gyneco-oncologist; NS, not significant; GIT, gastro-intestinal tract; CCR, completeness of cytoreduction; CRS, cytoreductive surgery; HIPEC, hyperthermic intraperitoneal chemotherapy.

**Table 3 jpm-11-01313-t003:** Post-operative outcomes of the patients who underwent CRS–HIPEC.

	Non-MDT, N = 33	MDT, N = 99	*p*-Value
MV support duration, days	0.5 ± 0.8	1.0 ± 1.6	NS
ICU admission duration, days	1.1 ± 2.0	2.0 ± 5.7	NS
Hospitalization, days	18.9 ± 13.0	25.3 ± 18.0	NS
Complications			NS
No	16 (48.5%)	54 (54.5%)	
Yes	17 (51.5%)	45 (45.5%)	
Complication grade			NS
None, 0	16 (48.5%)	54 (54.5%)	
Minor, 1–2	11 (33.3%)	27 (27.2%)	
Major, 3–4	4 (12.1%)	14 (14.1%)	
Hospital mortality, 5	2 (6.1%)	4 (4.0%)	
Type of major complications			
Cardiovascular			NS
No	32 (97.0%)	96 (97.0%)	
Yes	1 (3.0%)	3 (3.0%)	
Pulmonary			NS
No	28 (84.8%)	82 (82.8%)	
Yes	5 (15.2%)	17 (17.2%)	
Intraabdominal infection			NS
No	29 (87.9%)	85 (85.9%)	
Yes	4 (12.1%)	14 (14.1%)	
Follow up status			
Alive without recurrence	18 (54.5%)	52 (52.5%)	NS
Alive with recurrence	9 (27.3%)	28 (28.3%)	NS
Death	6 (18.2%)	19 (19.2%)	NS
Overall survival, months			
Mean ± SD (95% CI)	33.0 ± 2.9 (27.4–38.7)	28.2 ± 2.1 (24.0–32.4)	NS
3-month-OS rate	90.5%	91.5%	NS
6-month-OS rate	81.3%	76.7%	NS
12-month-OS rate	81.3%	74.1%	NS

Abbreviation: MV, mechanical ventilator; NS, not significant; ICU, intense care unit; OS, overall survival; SD, standard deviation; CI, confidence interval.

**Table 4 jpm-11-01313-t004:** Univariate and multivariate analyses of significantly predictive factors on appearance of post-operative complications for the patients after CRS–HIPEC.

	Univariate ^a^			Multivariate		
	HR	95% CI	*p*-Value	HR	95% CI	*p*-Value
Age						
≤65	1			1		
>65	2.83	1.01–7.98	0.049	3.19	0.96–10.61	0.058
Cancer type						
Ovarian	1					
Colorectal	1.26	0.49–3.25	NS			
Appendiceal	1.61	0.42–6.17	NS			
Gastric	1.61	0.43–5.59	NS			
Others	3.42	0.56–5.71	0.015			
Cancer status						
Primary	2.08	1.04–4.16	0.039			
Recurrent	1					
PCI class						
I (0–9)	1					
II (10–19)	2.78	1.08–7.18	0.035			
III (20–29)	3.45	1.28–9.32	0.015			
IV (30–39)	3.20	0.78–13.14	NS			
Operation time, hours						
≤12	1			1		
>12	3.05	1.37–6.83	0.007	3.54	1.33–9.43	0.011
Completeness of CRS						
CCR 0–1	2.30	1.02–5.22	0.046	3.48	1.09–11.05	0.035
CCR 2–3	1			1		
GIT anastomosis						
No	1			1		
Yes	2.46	1.21–4.98	0.013	2.58	1.01–6.55	0.047
MDT approach						
No	1.28	0.58–2.81	NS			
Yes	1					

Abbreviation: HR, hazard ratio; CI, confidence interval; NS, not significant; ECOG, Eastern Cooperative Oncology Group; PCI, Peritoneal Cancer Index; MDT, multidisciplinary team; CT, chemotherapy; GIT, gastrointestinal tract. ^a^ Following factors were calculated in UV: Gender, cancer types, ECOG, smoke, alcohol, diabetes, hypertension, abdominal operation history, co-malignancy, viral hepatitis, heart disease, pre-CRS CT, severity of clinical symptoms, operation method, blood loss amount, intraoperative blood transfusion, multi-visceral resection, and creation of enterostomy; only significant results are shown in this table.

**Table 5 jpm-11-01313-t005:** Univariate and multivariate analyses of significantly prognostic factors on survival for the patients after CRS–HIPEC by cox regression.

	Univariate ^a^			Multivariate		
	HR	95% CI	*p*-Value	HR	95% CI	*p*-Value
Age						
≤65	1			1		
>65	4.03	1.63–9.94	0.003	4.58	1.16–18.10	0.030
Gender						
Male	2.35	1.03–5.36	0.042			
Female	1					
Cancer type						
Ovarian	1					
Colorectal	2.97	0.86–10.28	0.087			
Appendiceal	3.96	0.95–16.57	0.060			
Gastric	9.44	2.42–36.84	0.001			
Others	3.95	1.20–13.01	0.024			
ECOG						
0	1					
1	3.78	1.24–11.51	0.019	1		
2	8.64	2.42–30.87	0.001	6.41	1.20–34.14	0.030
Completeness of CRS						
CCR 0–1	1			1		
CCR 2–3	3.99	1.77–8.98	0.001	2.79	1.04–8.27	0.048
Complication grade (0–4) ^b^						
None, 0	1					
Minor, 1–2	2.90	0.92–9.14	0.070			
Major, 3–4	4.48	1.30–15.52	0.018			
MDT						
No	1					
Yes	1.19	0.44–3.20	NS			

Abbreviation: NLR, neutrophil-to-lymphocyte ratio; PLT, platelet; ALB, albumin; CRS, cytoreductive surgery; GIT, gastrointestinal tract; MDT, multidisciplinary team; NS, not significant. ^a^ Following factors were calculated in UV: Smoke, alcohol, diabetes, hypertension, abdominal operation history, pre-CRS CT, primary or recurrent cancer, PCI class, severity of clinical symptoms, operation time, blood loss amount, intraoperative blood transfusion, multiple visceral resections, GIT anastomosis, and creation of enterostomy; only significant results are shown in this table. ^b^ Surgical mortalities are excluded.

## Data Availability

The data presented in this study are available on request from the corresponding author.

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
