# Peer review of "The Impact of Multidisciplinary Team Approach on Cytoreductive Surgery with Hyperthermic Intraperitoneal Chemotherapy for Peritoneal Carcinomatosis"

_jpm, 2021, doi:10.3390/jpm11121313_

Round 1

Reviewer 1 Report

Thank you for asking me to review this paper.

The authors are sharing their experience with MDT in peritoneal cancers cytoreductive surgery. I have major concerns about the methodology.

1- the heterogeneity of different cancers from different primaries (gastric, ovarian, colorectal etc) in one study may mislead as there is selection bias. The peritoneal cancer MDT model needs to be first proven to be the correct model rather than having a specialty led model such as Gynae Oncology MDT for example. In my opinion for example a specialty led MDT works better. 

2- There is no discussion of other important prognostic factors such as gene mutations, type of chemotherapy used, other target immunotherapy.

3- The selection bias may exist in assuming it is the MDT effect, it simply could be a centralizing effect where less complications and higher resectibility are due to more subspecialised surgeons and more trained

4- I could not understand a prophylactic use of HIPEC

Author Response

Response to reviewers’ comments

We are very grateful for the reviews provided by the editors and reviewers of this manuscript. The comments are encouraging and important. Please see below, our detailed responses to individual comments. All page numbers refer to the manuscript file with tracked changes.

Reviewer 1: The authors are sharing their experience with MDT in peritoneal cancers cytoreductive surgery. I have major concerns about the methodology.

1-1 the heterogeneity of different cancers from different primaries (gastric, ovarian, colorectal etc) in one study may mislead as there is selection bias. The peritoneal cancer MDT model needs to be first proven to be the correct model rather than having a specialty led model such as Gynae Oncology MDT for example. In my opinion for example a specialty led MDT works better. 

Author’s response:

We are grateful for this comment as it points to an important rationale of this study. An MDT model aims to bring together the range of specialists required to discuss and agree with treatment recommendations due to the increasing complexity of diagnostic and treatment decision-making. In this study, An MDT peritoneal malignancy care is defined using integrated resources of core (mandatory) and extended (recommended) elements. Nevertheless, a specialist team approach to peritoneal malignancy management may lack randomized controlled trial evidence of effectiveness, but is considered superior based upon both clinical consensus and research evidence. We agree with the reviewer that it may now be important to reevaluate the structure and models of MDT-work to determine how the best teams work, what comprises the best team nucleus and leadership styles, the best methods to allow shared learning, self-assessment, and feedback, and the most appropriate outcomes to enable more accurate evaluation of MDT care.

1-2 There is no discussion of other important prognostic factors such as gene mutations, type of chemotherapy used, other target immunotherapy.

Author’s response:

We thank and agree with the reviewer for this important comment. The concept of an MDT approach peritoneal malignancy care was widely accepted in the institution after a prior research from our allied institution, the Chiayi branch of the Chang-Gung Memorial Hospital, reported a higher complete cytoreduction rate and lower major complication rate of CRS-HIPEC under MDT guidance (Wang, T.Y.; Chen, C.Y.; Lu, C.H.; Chen, M.C.; Lee, L.W.; Huang, T.H.; Hsieh, M.C.; Chen, C.J.; Yu, C.M.; Chuang, H.C.; Liao, T.T.; Tseng, C.W.; Huang, W.S., Cytoreductive surgery with hyperthermic intraperitoneal chemotherapy for peritoneal malignancy: preliminary results of a multi-disciplinary teamwork model in Asia. Int J Hyperthermia 2017, 1-8.). This study aimed to determine the impact of the MDT approach on CIRS-HIPEC effectiveness, safety, and outcomes and focused on our first 3-year experience so that the forementioned other important prognostic factors were out of the scope. However, we would like to share an organizing study regarding the prediction of early mortality (died within 6 months after CRS/HIPEC surgery) with the reviewer. Other than poor performance and incomplete CRS, higher pre-operative NLR and NLR change other than persistent low values seem to be significant risks. Each of the controlled predictors is independent but related to others. High NLR was associated with several negative clinicopathological factors including older age, poor ECOG, insufferable clinical symptoms, progressive PCI classes, incomplete CRS, and poorly differentiated histology. It indicates that higher baseline NLR not only correlates with more aggressive tumor burden but also reflects patient’s physical and anti-cancer ability, both are vital in cancer treatment. A hypothesized theory emphasizes that inflammatory changes may perpetuate a pro-tumor microenvironment (Guthrie, G.J., et al., The systemic inflammation-based neutrophil-lymphocyte ratio: experience in patients with cancer. Crit Rev Oncol Hematol, 2013. 88(1): p. 218-30.) and be responsible for pro-tumor microenvironment. NLR more likely reflects a systemic status, especially for metastatic cancerous condition (Aino, H., et al., Clinical characteristics and prognostic factors for advanced hepatocellular carcinoma with extrahepatic metastasis. Mol Clin Oncol, 2014. 2(3): p. 393-398.). The awareness has been spreading about the effect of immune reaction and systemic inflammation (Mantovani, A., et al., Cancer-related inflammation. Nature, 2008. 454(7203): p. 436-44.) on cancer progression and even tumor metastasis assistance (Diakos, C.I., et al., Cancer-related inflammation and treatment effectiveness. Lancet Oncol, 2014. 15(11): p. e493-503.) by pro-tumor chemokines and matrix-degrading proteins. When malignancy grows and progresses and gives rise to tissue damage while invading local tissue. Acute-phase response reacts and tries to get rid of unusual processes, therefore contributing a systemic inflammatory response with neutrophilia and lymphopenia. Neutrophilia relates with increased cytokines (IL-1, 6), cytotoxic mediators, and vascular endothelial growth factors, and subsequently creates carcinogenesis, facilitates tumor invasion, and finally causes therapeutic resistance (Gungor, N., et al., Genotoxic effects of neutrophils and hypochlorous acid. Mutagenesis, 2010. 25(2): p. 149-54.). Lymphopenia represents a setback in adaptive immune surveillance and may inhibit malignant cell apoptosis, causing immune escape (Kim, R., M. Emi, and K. Tanabe, Cancer immunoediting from immune surveillance to immune escape. Immunology, 2007. 121(1): p. 1-14.). In tumor immunology, when neutrophilia combines with lymphopenia, a high NLR status, it appears for not only aggressive tumor biology but also compromised host anti-cancer ability (Schreiber, R.D., L.J. Old, and M.J. Smyth, Cancer immunoediting: integrating immunity's roles in cancer suppression and promotion. Science, 2011. 331(6024): p. 1565-70.). We are now constructing an NLR-based predict model (enrolled all our 6-year CRS/HIPEC cases) and planning to apply it on specific cancer (gastric cancer) and correlate with post-operative adjuvant or palliative chemotherapy/immunotherapy to draw a valid conclusion.

1-3 The selection bias may exist in assuming it is the MDT effect, it simply could be a centralizing effect where fewer complications and higher resectability are due to more subspecialised surgeons and more trained

Author’s response:

We thank the reviewer for this comment. Clinical care has become more complex and specialized, forcing medical staff to deliver complicated health services and to quickly learn new skills. CRS/HIPEC surgery involves complex diagnostic and perioperative decision-making and patient caring, as well as the task of possible multi-organ resection, have forced medical staff into the multidisciplinary approach. We believe that more subspecialised and more well-trained surgeons are helping to break down the hierarchy and centralized power of a single surgical team, giving more leverage to co-surgeons and producing a higher level of work.

1-4 I could not understand a prophylactic use of HIPEC

Author’s response:

We thank the reviewer for this comment.

The benefit of prophylactic HIPEC is mainly evidenced by high risk gastric (Brenkman HJF, Päeva M, van Hillegersberg R, Ruurda JP, Haj Mohammad N. Prophylactic Hyperthermic Intraperitoneal Chemotherapy (HIPEC) for Gastric Cancer-A Systematic Review. J Clin Med. 2019 Oct 15;8(10):1685. doi: 10.3390/jcm8101685. PMID: 31618869; PMCID: PMC6832700.), appendiceal neoplasms (Tuvin D, Berger Y, Aycart SN, Shtilbans T, Hiotis S, Labow DM, Sarpel U. Prophylactic hyperthermic intraperitoneal chemotherapy in patients with epithelial appendiceal neoplasms. Int J Hyperthermia. 2016 May;32(3):311-5. doi: 10.3109/02656736.2016.1152514. Epub 2016 Apr 6. PMID: 27050712) and colon cancers. (Arjona-Sánchez A, Barrios P, Boldo-Roda E, Camps B, Carrasco-Campos J, Concepción Martín V, García-Fadrique A, Gutiérrez-Calvo A, Morales R, Ortega-Pérez G, Pérez-Viejo E, Prada-Villaverde A, Torres-Melero J, Vicente E, Villarejo-Campos P, Sánchez-Hidalgo JM, Casado-Adam A, García-Martin R, Medina M, Caro T, Villar C, Aranda E, Cano-Osuna MT, Díaz-López C, Torres-Tordera E, Briceño-Delgado FJ, Rufián-Peña S. HIPECT4: multicentre, randomized clinical trial to evaluate safety and efficacy of Hyperthermic intra-peritoneal chemotherapy (HIPEC) with Mitomycin C used during surgery for treatment of locally advanced colorectal carcinoma. BMC Cancer. 2018 Feb 13;18(1):183. doi: 10.1186/s12885-018-4096-0. PMID: 29439668; PMCID: PMC5812226.)

Reviewer 2 Report

Minor comments:

Line 22: remove 'exactly' before 99

Line 81-83: 'Change every now and then' to 'ad-hoc depending on clinical findings'

Line 101: remove 'On the other hand'

Line 113: consider 'If it is indicated on symptomatic, clinical or serological evaluation'

Line 153 and Line 178-183: I don't understand the rationale behind 'prophylactic or palliative CRS-HIPEC' procedure. This is a radical treatment for patients with PC and would not usually be indicated by palliative or prophylactic indications? Authors should clarify what is intended in these circumstances.

Line 263-264 starting interestingly. Unclear the meaning of general surgeons role as multiplex?

Line 313: Change 'insulting' to 'associated with'

Author Response

Response to reviewers’ comments

We are very grateful for the reviews provided by the editors and reviewers of this manuscript. The comments are encouraging and important. Please see below, our detailed responses to individual comments. All page numbers refer to the manuscript file with tracked changes.

Reviewer 2:

2-1 Minor comments as followings: 

Author’s response:

We thank the reviewer for this comment. We have made all necessary changes as suggested and re-uploaded the revised manuscript.

Line 22: remove 'exactly' before 99

Line 81-83: 'Change every now and then' to 'ad-hoc depending on clinical findings'

Line 101: remove 'On the other hand'

Line 113: consider 'If it is indicated on symptomatic, clinical or serological evaluation'

Line 313: Change 'insulting' to 'associated with'

2-2 Line 153 and Line 178-183: I don't understand the rationale behind 'prophylactic or palliative CRS-HIPEC' procedure. This is a radical treatment for patients with PC and would not usually be indicated by palliative or prophylactic indications? Authors should clarify what is intended in these circumstances.

Author’s response:

We thank the reviewer for this comment. Complete cytoreduction followed by HIPEC improves survival in patients with peritoneal carcinomatosis. However, CRS/HIPEC remains one of the most morbid treatments offered for advanced cancers. Patients with involvement of massive regions of the abdominal cavity, or predictable incomplete cytoreduction, had still a poor prognosis. Therefore, the potential role of "palliative CRS/HIPEC" in the management of peritoneal carcinomatosis is being raised, e.g., bypass surgery or creation of stoma to restore gastrointestinal tract continuity, resect organs that may cause clinical symptoms, and peritonectomy and HIPEC procedure for possible ascites control. Given the limited survival benefit expected after CRS/HIPEC, understanding the impact of the treatment on quality of life needs to be an essential part of the decision to proceed and is critical to optimizing recovery afterwards.

The benefit of prophylactic HIPEC is mainly evidenced by high risk gastric (Brenkman HJF, Päeva M, van Hillegersberg R, Ruurda JP, Haj Mohammad N. Prophylactic Hyperthermic Intraperitoneal Chemotherapy (HIPEC) for Gastric Cancer-A Systematic Review. J Clin Med. 2019 Oct 15;8(10):1685. doi: 10.3390/jcm8101685. PMID: 31618869; PMCID: PMC6832700.), appendiceal neoplasms (Tuvin D, Berger Y, Aycart SN, Shtilbans T, Hiotis S, Labow DM, Sarpel U. Prophylactic hyperthermic intraperitoneal chemotherapy in patients with epithelial appendiceal neoplasms. Int J Hyperthermia. 2016 May;32(3):311-5. doi: 10.3109/02656736.2016.1152514. Epub 2016 Apr 6. PMID: 27050712) and colon cancers. (Arjona-Sánchez A, Barrios P, Boldo-Roda E, Camps B, Carrasco-Campos J, Concepción Martín V, García-Fadrique A, Gutiérrez-Calvo A, Morales R, Ortega-Pérez G, Pérez-Viejo E, Prada-Villaverde A, Torres-Melero J, Vicente E, Villarejo-Campos P, Sánchez-Hidalgo JM, Casado-Adam A, García-Martin R, Medina M, Caro T, Villar C, Aranda E, Cano-Osuna MT, Díaz-López C, Torres-Tordera E, Briceño-Delgado FJ, Rufián-Peña S. HIPECT4: multicentre, randomized clinical trial to evaluate safety and efficacy of Hyperthermic intra-peritoneal chemotherapy (HIPEC) with Mitomycin C used during surgery for treatment of locally advanced colorectal carcinoma. BMC Cancer. 2018 Feb 13;18(1):183. doi: 10.1186/s12885-018-4096-0. PMID: 29439668; PMCID: PMC5812226.)

We added the following sentences to give explanatory information in the Methods and Results sections.

Page 02, the fifth paragraph (Materials and methods):

Palliative CRS in the management of peritoneal carcinomatosis is mainly for patients with massive tumor burden or extensive abdominal cavity involvement, or predictable incomplete cytoreduction. Prophylactic HIPEC applies for non-metastatic cancer patients at high risk of tumor recurrence after curative-intent surgery.

Page 04, the first paragraph (Results):

Curative-intent CRS-HIPEC procedures were performed for 111 (84.1%) patients, palliative operations were done for 18 (13.6%; 3 ovarian cancer, 4 colorectal cancer, 2 appendiceal cancer, and 4 with other primary malignancies) patients, and 3 (2.3%; 2 gastric cancer and 1 gallbladder cancer) cases underwent prophylactic operations.

2-3 Line 263-264 starting interestingly. Unclear the meaning of general surgeons’ role as multiplex?

Author’s response: We thank the reviewer for this comment. We modified the following sentences.

Page 10, the first paragraph (Discussions):

Interestingly, general surgeons were involved in diverse cancer types than the other two as procedure initiators in this study.

Reviewer 3 Report

Congratulations on your paper, it is truly a very interesting subject for surgeons and other medical professionals. 

However, I would like you to write a little bit more about why the non-MDT cases were treated as such (you only wrote "treatment process does not conform to defined regulations, 83 then the case would be addressed using the non-MDT approach" which I find a bit too confusing as to the methods of investigation).

Another suggestion would be to rearrange Tables 1-3 and use indetations instead of slashes to divide certain subgroups - it is a little difficult to follow when reading and it's a shame considering the information is quite interesting. 

Thank you for your paper and best of luck with your future research!

Author Response

Response to reviewers’ comments

We are very grateful for the reviews provided by the editors and reviewers of this manuscript. The comments are encouraging and important. Please see below, our detailed responses to individual comments. All page numbers refer to the manuscript file with tracked changes.

Reviewer 3: Congratulations on your paper, it is truly a very interesting subject for surgeons and other medical professionals. 

3-1 I would like you to write a little bit more about why the non-MDT cases were treated as such (you only wrote "treatment process does not conform to defined regulations, 83 then the case would be addressed using the non-MDT approach" which I find a bit too confusing as to the methods of investigation).

Author’s response:

We appreciate the comment from the reviewer. An MDT peritoneal malignancy care is defined using integrated resources of core (mandatory) and extended (recommended) elements. We added the following sentences to give explanatory information in the Materials and Method section.

Page 02, the second paragraph (Materials and Methods):

However, if the treatment process does not conform to defined regulations (e.g., the most responsible physician does not seek for an MDT approach or the MDT decision is discordant), then the case would be addressed as the non-MDT approach.

3-2 Another suggestion would be to rearrange Tables 1-3 and use indetations instead of slashes to divide certain subgroups - it is a little difficult to follow when reading and it's a shame considering the information is quite interesting. 

Author’s response:

We thank the reviewer for this comment. We have made all necessary changes and re-uploaded the revised manuscript.

Round 2

Reviewer 1 Report

Authored responded to comments well. However, I have only one minor comment

The authors could state that the MDT concept they have for peritoneal cancer is one way of MDT, the other way is subspec used Gynae oncology MDT, or colorectal MDT or gastric MDT

Treating all peritoneal cancers by one team is something not common in countries such as ImuK or USA where each cancer whether peritoneal or not is treated by the subspecialist team in that particular cancer

so ovarian cancer peritonectomy is done by Gynae oncology sugeons

gastric is by upper GI

colon by lower GI

etc

If authors mention this clearly they will show good undertow the global variation

Author Response

Response to reviewers’ comments

We are very grateful for the reviews provided by the editors and reviewers of this manuscript. The comments are encouraging and important. Please see below, our detailed responses to individual comments. All page numbers refer to the manuscript file with tracked changes.

Reviewer 1: Authored responded to comments well. However, I have only one minor comment. The authors could state that the MDT concept they have for peritoneal cancer is one way of MDT, the other way is subspec used Gynae oncology MDT, or colorectal MDT or gastric MDT. Treating all peritoneal cancers by one team is something not common in countries such as ImuK or USA where each cancer whether peritoneal or not is treated by the subspecialist team in that particular cancer. So ovarian cancer peritonectomy is done by Gynae oncology sugeons. gastric is by upper GI. colon by lower GI, etc.

If authors mention this clearly they will show good undertow the global variation

Author’s response:

We are grateful for this comment as it points to an important emphasis of this study.

We added the following sentences to give explanatory information in the discussions section.

Page 10, the first paragraph (Discussions):

Our CRS-HIPEC MDT team initiator could belong to a subspecialty which presents as the domination of the CRS process rather than adopting a subspecialist team in particular cancer. The concept is to treat all peritoneal cancers by the most advantageous MDT team approach, and to bring together the range of specialists is nevertheless a necessity for a comprehensive treatment plan for not only intraoperative assistance but also postoperative adjuvant therapy.
